# Classification of images of bee pollen according to their producers

**Juan D. Leal Campuzano**[1,2], **Carlos A. Martínez Niño**[3], **Francisco A. Gómez Jaramillo** [4]*

**1** Dirección de Investigación y Desarrollo/Corporación Colombiana de Investigación Agropecuaria - AGROSAVIA, Madrid, Cundinamarca, Colombia, **2** Facultad de Sistemas/Escuela Tecnológica Instituto Técnico Central, Bogotá, Colombia, **3** Departamento de Producción Animal, Facultad de Medicina Veterinaria y de Zootecnia, Universidad Nacional de Colombia, Bogotá, Colombia, **4** Departamento de Matemáticas, Facultad de Ciencias, Universidad Nacional de Colombia, Bogotá, Colombia

* fagomezj@unal.edu.co

**Data availability statement:** The data is stored in: https://www.kaggle.com/datasets/juanleal0317/pollen-samples-from-a-boyaca-region.

## Abstract

The food industry is witnessing a growing interest in pollen due to its nutritional and energy composition. Consumers of bee pollen are increasingly eager to learn about the origins of the products they purchase. Establishing the geographical origin and the producer of pollen can enhance the product's value and meet consumer demands for transparency in the supply chain. This article presents a novel approach for the classification of images of bee pollen according to their producers using digital images and machine learning. The study focuses on pollen collected from various beekeepers in the Boyacá region of Colombia. A standardized image acquisition process was employed to capture macroscopic images of the pollen samples. These images were then analyzed to extract color information, and machine learning models were trained to predict the producer of the pollen based on its color characteristics. The results demonstrate that the proposed approach can effectively determine the producer of pollen samples based on their color information. The model achieved an accuracy rate of 85% in associating pollen samples with their respective beekeepers. This outcome has significant implications for traceability and transparency in the bee pollen industry, offering a cost-effective and accessible method to verify the product's origin.

## Introduction

The food industry is increasingly interested in pollen due to its nutritional and energy composition [1]. Bee pollen has also piqued the interest of the pharmaceutical and personal care industries [2]. Additionally, consumers are increasingly eager to learn more about the origin and production processes of the products they purchase [3]. For example, the *seal of origin*, which identifies unique products from specific regions, demonstrates how geographical product origin can be relevant to consumers [4]. In recent years, various beekeeping-based products have earned these seals of origin [5,6]. However, consumer demand for transparent supply chain practices continues to challenge the bee pollen industry to provide visibility and disclose this information to consumers [7].

**Funding:** The author(s) received no specific funding for this work.

The development of beekeeping has positive agricultural impacts [8]. For instance, approximately 70% of crops experience increased production when pollinators visit their flowers [9]. Worldwide, about 12 pollinator species are commercialized, with *Apis mellifera* and *bombus terrestris* being the most widespread [10]. Most apiaries primarily focus on honey production using these pollinators; however, in regions where environmental conditions support diverse pollen flora, such as in some developing countries [11], pollen collection and commercialization have become significant economic activities [11]. Nonetheless, there is a need for improved mechanisms to add value to the pollen production chain, particularly in developing countries [11].

One way to meet customer requirements and enhance pollen production and commercialization is by linking the product to its geographical origin [12]. Associating a product with its origin connects information about the product, including its reputation, quality, and nutritional composition, with the production location and the producer, ensuring high-quality products and increasing their economic value [4]. However, implementing this added value mechanism requires addressing the challenge of determining the geographical origin of pollen, i.e., determining the origin of a given pollen sample [13,14].

It is worth recalling that honeybee foraging behavior, particularly in *Apis mellifera*, is characterized by floral constancy, whereby worker bees consistently visit the same plant species once selected. This behavior, reported by [15–17], results in minimal variability in the botanical origin of collected pollen over time, thus reducing color variation attributable to plant species, particularly those that provide the most nutrients for the hive [18]. Consequently, pollen may potentially encode geographic information through its botanical origin. Based on this observation, various methods have been proposed to determine the geographical origin of pollen. Palynology and bromatology are the most widely used methods for establishing pollen's geographical origin [19]. However, these methods are time-consuming and require specialized training, which is not readily available in productive areas of developing countries [20]. Physicochemical processes are also used for pollen origin characterization [21], but these methods require expensive and specialized equipment that is inaccessible to many participants in the pollen production chain [1]. More recently, computer vision has emerged as an alternative for assessing certain pollen properties, such as its composition and the color characteristics [22–24].

Color information has been previously explored in pollen characterization. For example, in [25], a method is proposed to characterize bee pollen using visual classification, microscopy, and vibrational spectroscopy. The authors used CIE Lab* values to measure color, and Fourier Transform-Raman spectroscopy to identify pigment differences, especially in carotenoids and flavonoids. A multivariate analysis helped distinguish pollen types, showing the method's value for quality control and authentication; in [26] bee pollen is examined from several regions in Germany, identifying 14 major European pollen types. Pollen was classified using light microscopy, and color data were analyzed with Gaussian Mixture Models. The results showed high color variation within species, often beyond human perception, highlighting the limitations of human color-based species identification. While these methods advance botanical description, they rely on microscopy or complex techniques and do not address the determination of the geographical origin. Complementary studies show that regional differences affect the physicochemical and nutritional properties of bee pollen [27,28]. In [28], color, pH, protein, and fatty acid profiles are linked to Brazilian regions. In [27], a location-dependent variation was found in protein, fiber, lipid, and phenolic content in *Melipona mandacaia* pollen. These results support the use of color metrics to infer geographical origin.

This work proposes a novel strategy for the classification of images of bee pollen according to their producers using digital images acquired under semi-controlled conditions and machine learning-based methods. Notably, once the producer is identified, its location can be linked to the geographical location, providing an alternative way to characterize the spatial pollen origin. The method links the color composition of macroscopic pollen images, i.e., images taken at a real-life scale, to the producer of the sample using machine learning. This approach is more accessible and cost-effective compared with previous methods that relied on specialized acquisition devices and highly trained experts. To achieve this, a color segmentation approach for pollen and a supervised machine-learning strategy for automatically determining the origin of apiary pollen were used. The primary contribution of this work is provide evidence, for the first time, that macroscopic pollen images can also offer information about color composition related to the producer and its geographical origin. This discovery opens up new possibilities for studying other pollen properties under simpler acquisition conditions, including pollen nutritional composition and other tasks required by the pollen production chain.

## Materials and methods

In Fig 1, we illustrate the proposed approach for the classification of macroscopic images of bee pollen according to their producers. The process begins with the collection and preparation of samples from a group of pollen producers. Subsequently, a standardized acquisition process generates a set of pollen sample images. Afterward, a preprocessing stage is employed to enhance image quality. Color characteristics then serve as a discriminative representation of each pollen type on the pollen spectrum. A machine learning classification model, based on the producer's location, is used to link the image's color information to the apiary where the pollen was produced. This model is evaluated by predicting the pollen producer from images acquired under the same conditions.

### Pollen samples

The samples were collected in 2021 from paramo and humid ecosystems during each flowering season to capture potential variations in bee preferences. This study involved beekeepers from various locations in the Márquez region, including the municipalities of Nuevo Colón, Turmequé, Viracachá, and Ciénega, and the Tundama region, encompassing the municipalities of Tibasosa, Belén, Santa Rosa de Viterbo, Duitama, Paipa, and Tuta, all located within the Boyacá department of Colombia. These municipalities are located in regions known for high pollen production.

For the sample acquisition, a comprehensive survey was first conducted to determine the flowering periods of the plant species present in the ecosystems relevant to the project. This information was essential for subsequently identifying the origin of the pollen collected by the colonies in the participating apiaries. A one-year collection calendar was developed in collaboration with local beekeepers and two local experts in bee production systems with field experience comprising four distinct pollen sampling periods. To set this number of visits, the producers and two local experts in bee production systems with field experience considered the variations in the flowering cycle of the region under study. It is worth mentioning that in tropical areas as the region under study, in contrast to other regions, there are no marked seasons, and consequently, most plants visited by bees blossom throughout the whole year [30]. This factor contributes to reducing the floristic variability along the year, contributing to minimizing the number of visits for collecting samples.

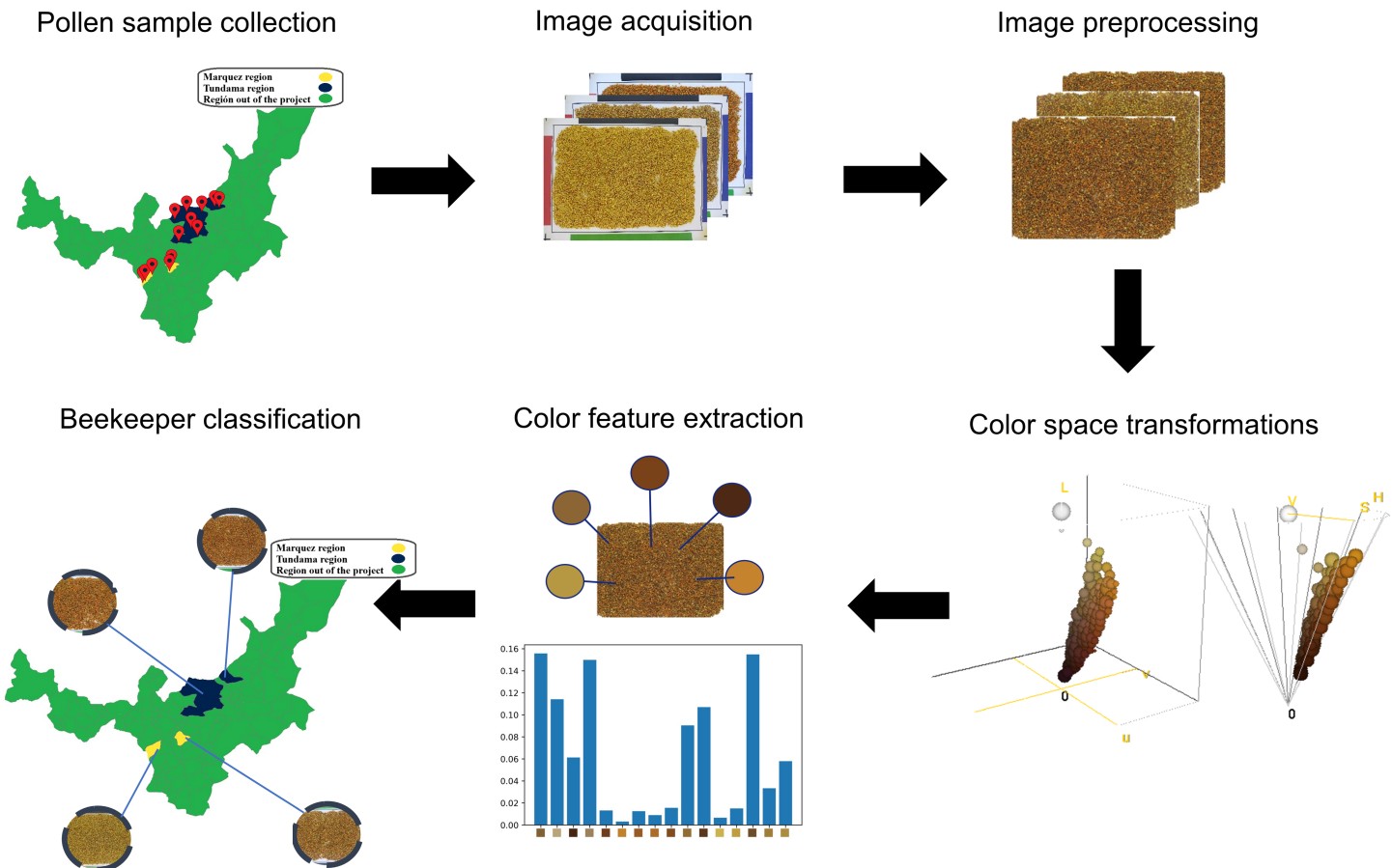

**Fig 1. Flowchart of the geographic origin classification process.** Identification of the geographic origin of the pollen begins with collecting a set of pollen samples from various regions. Macroscopic pollen images are acquired under standardized conditions, as mentioned in the sample image acquisition subsection. A preprocessing process is applied to enhance the images, followed by the extraction of color information from the images. A machine learning-based classification method is then used to establish the connection between color information and the pollen producer. The map was created by the authors using Geopandas in Python version 3 and the data were taken from Overpass Turbo [29].

A total of twenty beekeepers were instructed to collect pollen samples biweekly during each period, resulting in six collections over a three-month span. In each collection round, five hives were randomly selected. To avoid repetition, no hive was to be sampled more than once per period. On average, 200 grams of pollen were collected per hive during each sampling event, and the samples were stored under refrigeration until retrieval. In each sampling period, 30 samples per beekeeper were expected. This procedure was repeated during all four periods to account for temporal variability; some challenges arose due to rainfall, which hindered pollen collection by the bees. As a result, in certain apiaries, only four of the six planned biweekly collections were completed, yielding 24 samples per period instead of the intended 30. The number of samples collected each time per producer is shown in Fig 2, which presents the distribution of the number of pollen samples per producer across all visits.

Honeybee foraging behavior, particularly in *Apis mellifera*, is characterized by floral constancy, whereby worker bees consistently visit the same plant species once selected. This behavior contributes to reducing the variability in the botanical origin of collected pollen over time [15,17,31], thereby reducing color variation attributable to plant species.

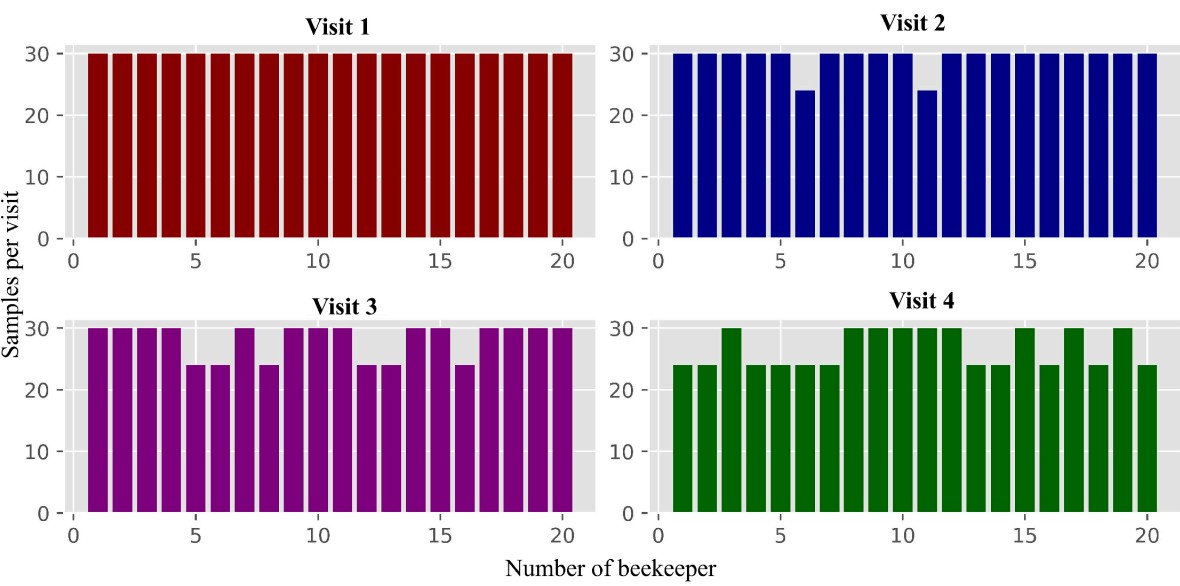

**Fig 2. Frequency distribution of pollen samples per producer across the different sampling visits.**

For controlling variability in pollen production, the producers involved in the study followed a standardized production protocol [32]. This protocol specified the drying method, with a temperature range of 50 to 60 degrees Celsius, using stainless steel machinery, and a drying duration of 12 to 24 hours to achieve a moisture content ranging from 5% to 8%. For this study, a drying time of 24 hours was applied to each sample, resulting in a final moisture content of 5%. Since the drying parameters were standardized, it is expected that the variability in color associated with this extrinsic characteristic will not have a significant impact. The pollen samples were maintained at a temperature of 55 $C^o$ to facilitate grain manipulation. For each sample, 100 grams of dried pollen were manually spread across a non-reflective template, ensuring a uniform distribution that avoided both clumping and excessive empty space. This process was repeated before capturing each of the four images per sample, reducing the influence of local dispersion artifacts and maximizing the representativeness of the sample's color composition.

### Sample image acquisition

Images were acquired under standardized conditions to reduce variability in illumination and preserve color information. Two separate image acquisitions, each consisting of two images, were performed at two different times by two researchers to assess the reliability of the approach. The acquisition protocol involved using a light-box to capture images of pollen samples under controlled lighting and geometric conditions at a macroscopic scale. The acquisition device was equipped with an LED-based lighting system and a matching interior cover designed to minimize reflections and reflectance artifacts in the acquired images. The dimensions of the device and the camera position were adjusted to optimize the field of view for the pollen sample located at the bottom. Additionally, the light-box featured a reference color mark with red, green, blue, and black guidelines placed on the box's base, which was later used for standardizing the colors in the images. The device utilized for capturing images of the pollen samples is presented in Fig 3.

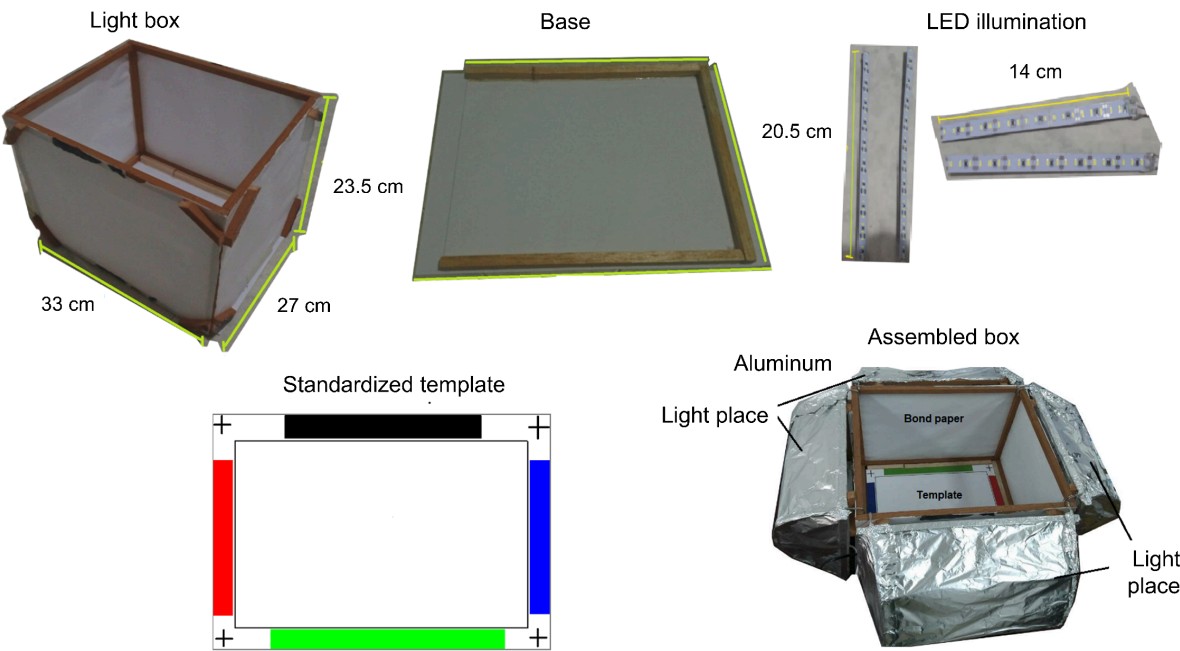

**Fig 3. Lightbox.** This instrument was designed to standardize the acquisition of pollen sample images at a macroscopic scale. The device was equipped with an LED-based illumination system to 12V and a matte interior and a lateral cover of Bond paper, which helped maintain consistent lighting and geometric conditions, The template made from a sheet of Propalcote paper, was designed to standardize color digitization

In this experiment, images were captured using a frontal cellphone Xiaomi REDMI Note camera with a resolution of 32 $Mp$, an aspect ratio $9:16$, high image quality enabled, an aperture of $f = 2$, a shutter speed of $s = 1/30$ and $ISO = 200$. Four images were acquired for each pollen sample. In Fig 4, we present an image corresponding to a pollen sample captured using the proposed method.

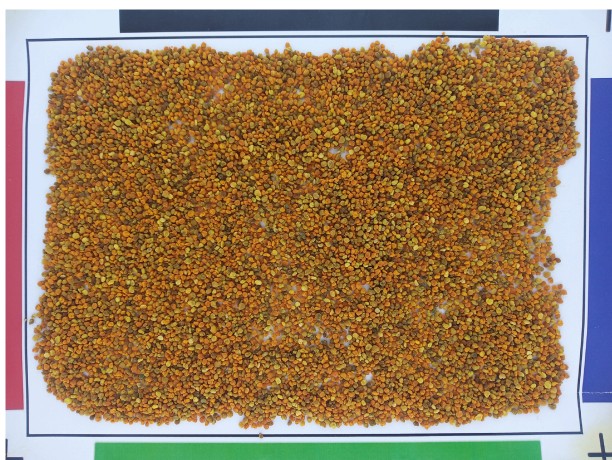

**Fig 4. Image pollen sample acquired under the proposed acquisition protocol.**

## Image preprocessing

An algorithm based on local entropy texture was used to segment the image area containing pollen grains [33]. Specifically, the entropy of the local histogram of gray levels in circular patches with a radius of five pixels was calculated. Subsequently, the Otsu binarization algorithm was applied to determine a threshold for separating the pollen (high entropy) from the background (low entropy) [34].

Additional post-processing was performed to eliminate any extra noisy pixels associated with small grain shadows in the image. To achieve this, the distance between the RGB color of each pixel and the diagonal of the RGB cube, which distinguishes different gray levels in an image within the RGB cube, was computed as follows:

$$\frac{\|U \times P\|}{\|U\|} \tag{1}$$

where $U = [1, 1, 1]$ represents the direction vector associated with the cube's diagonal line, and $P$ represents the RGB color triple. Once the distances for all the pixels were calculated, pixels with distances less than one standard deviation from the cube's diagonal line were considered part of the region of interest. In Fig 5, we present the cleaning process applied to a pollen image.

## Color space transformation

Color serves as a natural perceptual marker for characterizing pollen. The chemical compounds responsible for pollen color include pigments such as flavonoids and carotenoids [22]. These compounds can vary in quantity and chemical interactions within each pollen type, resulting in distinct color variations. Consequently, a relationship exists between the color of the pollen and the species from which it originates [35]. This study leverages the color information of pollen grain images to create a distinctive feature associated with the geographical location.

To capture the color characteristics associated with each pollen producer, the colors initially represented in the RGB color space were transformed into more suitable color representations. A color space is a multidimensional representation of the color spectrum [36]. For this study, two color spaces were considered for characterization: HSV (Hue, Saturation, and Value) and $L^*u^*v^*$ of the CIE (Commission Internationale de l' Éclairage) [36].

The HSV color space is designed to separate the luma (value) from the color information (hue and saturation). This representation is invariant to variations in light intensities and shadows, which can otherwise distort the color information, allowing for a clear description

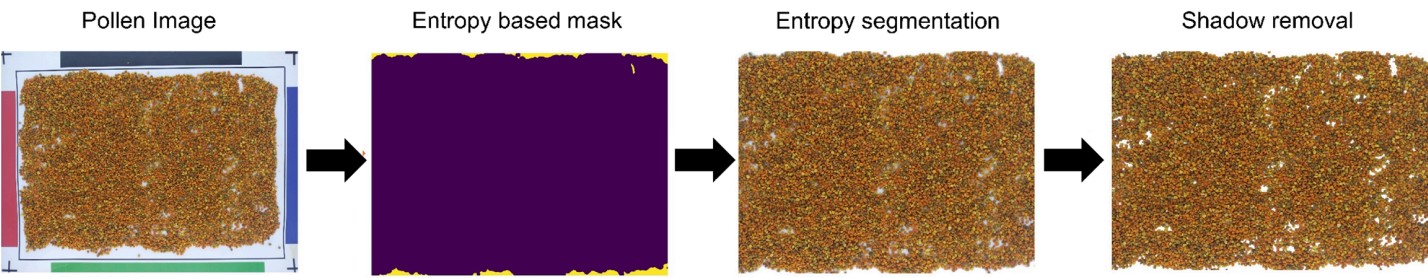

**Fig 5. Preprocessing of images.** Each image of pollen sample was processed from digitization (left) to shadow removal (right).

of the color characteristics in the pollen grains [37]. The conversion of colors for each pixel from the RGB space to the HSV space was carried out as follows:

$$H = \arctan\left(\frac{\sqrt{3}(G-B)}{2R-G-B}\right) \tag{2}$$

$$S = \begin{cases} 0 & if \ C = 0 \\ \frac{C}{V} & otherwise. \end{cases} \tag{3}$$

$$V = \max(R, G, B) \tag{4}$$

where $C = \max(R, G, B) - \min(R, G, B)$.

The $L^*u^*v^*$ color space offers a perceptually uniform color characterization, meaning that color differences computed in this space are proportional to how humans perceive these differences [37–39]. In the $L^*u^*v^*$ color space, colors that appear similar in pollen are positioned close to each other. At the same time, pixels perceived as different by human perception are positioned farther apart. This representation enhances the intraclass/interclass variability for similar and different colors, respectively. The transformation of pixels from RGB to the $L^*u^*v^*$ color space was carried out as follows:

$$\begin{pmatrix} X \\ Y \\ Z \end{pmatrix} = \begin{pmatrix} 0.607 & 0.174 & 0.2 \\ 0.299 & 0.587 & 0.114 \\ 0 & 0.066 & 1.116 \end{pmatrix} \begin{pmatrix} R \\ G \\ B \end{pmatrix} \tag{5}$$

followed by a nonlinear transformation that takes into consideration the standard values for the white color $(Y_0, u_0, v_0)$ according to the CIE, as follows:

$$L = \begin{cases} 116\sqrt[3]{Y} - 16 & for \ Y > 0.008856 \\ 903.3 \cdot Y & for \ Y \leq 0.008856 \end{cases} \tag{6}$$

$$u^* = 13L\left(\frac{4X}{X + 15Y + 3Z} - u_0\right) \quad where \quad u_0 = 0.19793943 \tag{7}$$

$$v^* = 13L\left(\frac{6Y}{X + 15Y + 3Z} - v_0\right) \quad where \quad v_0 = 0.46831096. \tag{8}$$

The original RGB images were characterized using four color representations: HSV, $L^*u^*v$, HS, and $^*u^*v$, to investigate the hypothesis that color is a highly informative feature for this task. It is worth noting that the last two representations only encode color information.

## Color feature extraction

Once suitable representation spaces had captured the pollen color information, a set of low-dimensional features was constructed for input into machine learning algorithms. To achieve this, a color quantization algorithm was applied to generate a compact set of representative colors for the pollen grains under study [40]. The $k$-means clustering algorithm was used to identify the most representative colors in the pollen images. To address the computational

challenge posed by the large number of pixels in all the pollen images, a *k*-means mini-batch algorithm was employed [41,42]. This algorithm divides the clustering process into smaller batches, improving processing times. The number of species identified in an independent palynological analysis conducted on the collected pollen samples determined the number of centroids for the *k*-means algorithm. The palynological study reported between 16 and 35 different species per sample. Therefore, the *k* parameter also ranged from 16 to 35. Each pixel was assigned to the nearest representative color determined by the *k*-means algorithm based on the minimum Euclidean distance. Subsequently, each image was represented by a *k*-dimensional vector containing the frequencies of occurrence of the *k* key colors within the image. This representation was calculated for all four color spaces under investigation. In Fig 6, we illustrate the different feature vectors obtained for the various color spaces computed for a sample.Although palynological analysis revealed that the number of floral types per producer ranged from 16 to 35, this variability was not used to define producer-specific color clustering. Instead, a fixed number of centroids was applied to all samples to ensure comparability and standardization. This value was selected after cross-validating performance across the tested range. Since the number of floral species may vary, the analysis of this parameter revealed that using 32 color centroids provided the best representation across all samples. This standardization ensured that each input vector had the same dimensionality, independent of regional species variations.

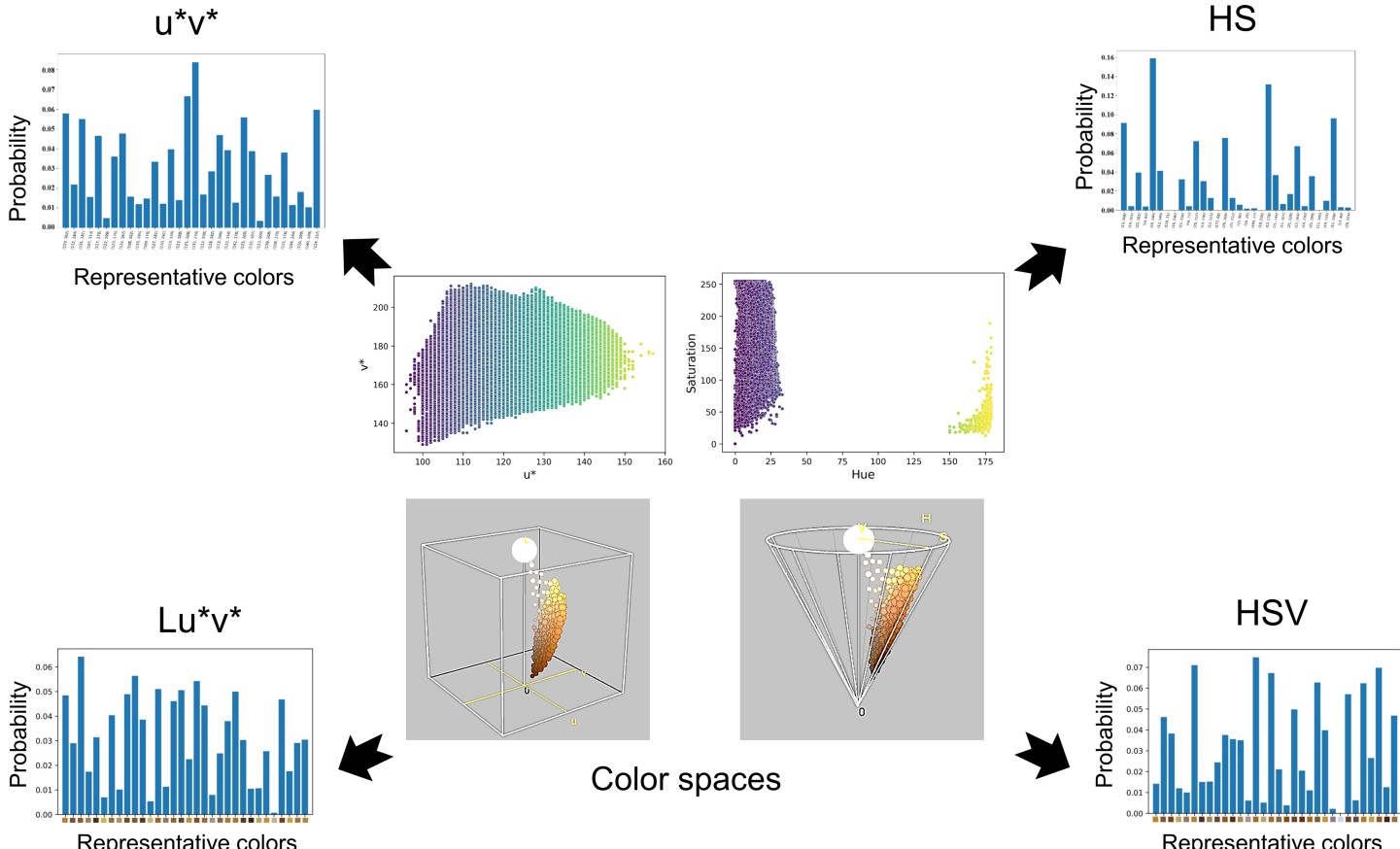

**Fig 6. Histograms representing characteristic vectors for each color space.**

## Classification of the pollen producer

Utilizing feature vectors that describe the color information of the pollen, a multiclass classification strategy based on machine learning was employed to determine the pollen's producer and, consequently, its geographical location. Specifically, a set of classifiers was created, with color feature vectors as input and the identification of the beekeeper as the output. Various multiclass classification strategies were explored, including support vector machines with polynomial kernels with degrees varying between 4 and 10, random forest with up to 200 trees in the forest, and a multilayer perceptron with varying numbers of neurons and different quantity of hidden layers [43]. To examine the temporal generalizability of the classification model, the test partition was divided into four subsets corresponding to each seasonal sampling visits. In a temporal testing setup, the model was evaluated on all four visits, simulating prediction on an unseen time window. This approach allowed us to study the model's ability to provide unbiased performance under temporal shifts in the pollen spectrum.

## Experimental evaluation

The experimental setting aimed to quantify the capacity of the proposed strategy to correctly determine the beekeepers using images of pollen samples acquired under the proposed protocol.

According to the methodological proposal, a total of 4,800 images were expected. However, due to practical difficulties related to adverse weather conditions in the field, the number of collected samples was reduced to 4,572. This reduction resulted in a slight class imbalance, later accounted for in the classifiers' training and validation stages. A total of 3,048 images were considered in the training and validation sets, and 1,524 pollen images in the test set to evaluate the classifier's performance. Each machine learning strategy was trained using the four different color representations, i.e., each color representation resulted in a different classifier. For each classifier, different hyperparameters (or tuning parameters) were also considered. A stratified cross-validation procedure selected the best classification model and the corresponding hyperparameters [44]. This data partition strategy is a variation of k-fold that returns stratified folds. In this case, ten folds were created to preserve the percentage of samples for each class. This strategy ensures the training and validation sets have the same proportions of beekeepers as in the original dataset.

The predictive performance was evaluated using the multiclass confusion matrix with accuracy, precision, recall, and F1-score as parameters. The macro-precision, macro-recall, and macro F1-score were also computed for the evaluation. The weighted-average F1 score, calculated as the weighted mean of all per-class F1-scores using the number of samples per class as weights, was used for model selection in the validation partition [45].

The repeatability, also called test–retest reliability, of the proposed method for the beekeeper association task was also studied [46]. In this case, test-retest reliability refers to the consistency of the proposed classification system when the same pollen sample is classified multiple times under the same conditions but at different points in time. This performance measurement assesses how well the classification results, i.e., the pollen producer assignment, can be reproduced if the test is repeated on a different pollen image sample. The goal is to ensure the beekeeper classification method is stable over time and yields similar results when the same data is classified again. For this evaluation, the test partition for each beekeeper was divided into two subsets taken at different times and containing the same number of images per beekeeper. Then, the trained classifier was used on the first data set to compute individual F1 scores for each beekeeper. Then, the second data set corresponding to a different point in time was used to compute these F1 scores as well. Finally, the repeatability between the first

and second datasets ($r_{1,2}$) was calculated as the Pearson correlation coefficient between the individual F1 scores obtained for each beekeeper in the two test partitions, as follows:

$$r_{1,2} = \frac{\sum_{i=1}^{n}(f_{i,1} - \bar{f}_1)(f_{i,2} - \bar{f}_2)}{\sqrt{\sum_{i=1}^{n}(f_{i,1} - \bar{f}_1)^2}\sqrt{\sum_{i=1}^{n}(f_{i,2} - \bar{f}_2)^2}}$$

where $f_{i,1}$ is the F1 individual score obtained for the $i$-th pollen producer on the first dataset, $f_{i,2}$ is the F1 score obtained for the $i$-th pollen producer on the second dataset, $\bar{f}_1$ is the mean of the F1 individual scores for all beekeepers in the first dataset, $\bar{f}_2$ is the mean of the F1 individual scores for all beekeepers in the second dataset and $n$ is the number of beekeepers.

The used code and the image database needed to reproduce results can be accessed in the GitHub repository and the Image Dataset respectively.

## Results

Some examples of beekeeper associations provided by the proposed model for pollen images are presented in Fig 7. Relevant regions are highlighted in blue and yellow. As shown, the proposed model correctly associates beekeepers with the majority of the pollen images used as input. However, the red arrow in the figure indicates an erroneous association between

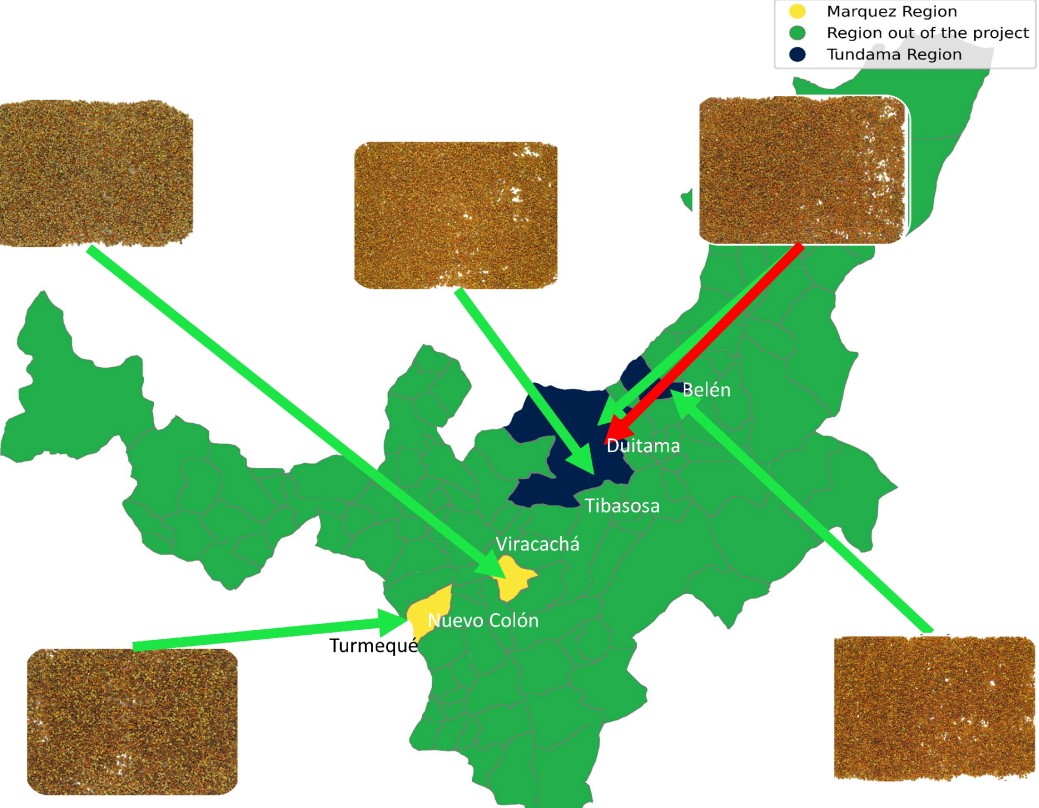

**Fig 7. Examples of beekeeper associations resulting from using pollen images.** Regions involved in the project (blue and yellow) and corresponding pollen images are shown. Green arrows indicate correct beekeeper identifications and the red arrow indicates an incorrect association. The map was created by the authors using Geopandas in Python version 3 and the data were taken from Overpass Turbo [29].

two pollen producers located in the municipality of Duitama (Boyacá). In this instance, the proposed model made an incorrect prediction, possibly due to the geographical proximity of both producers.

The confusion matrix obtained by applying the proposed strategy to a test partition of twenty pollen producers is shown in Fig 8. The municipalities where the beekeepers are located are listed in the rows of the first column of the figure. These results correspond to the model with the highest performance in the weighted-average F1-score, specifically a support vector machine (SVM) trained using a feature vector consisting of 32 reference colors resulting from clustering in the *uv* color space. The identified model achieved an accuracy of 85% in the beekeeper association task using pollen images.

As observed, the diagonal of the confusion matrix demonstrates that the proposed approach performs well. Notably, the method resulted in a macro-average recall of 84%, with individual recalls exceeding 80% for most beekeepers (15 out of 20), indicating the model's capability to identify instances of specific beekeepers correctly. Similarly, the macro-average precision was 85%, with individual precisions exceeding 80% for 17 out of 20 beekeepers, highlighting the model's capacity to identify only the pollen samples associated with particular beekeepers correctly. Finally, the total macro F1-score was also 85%, indicating a good

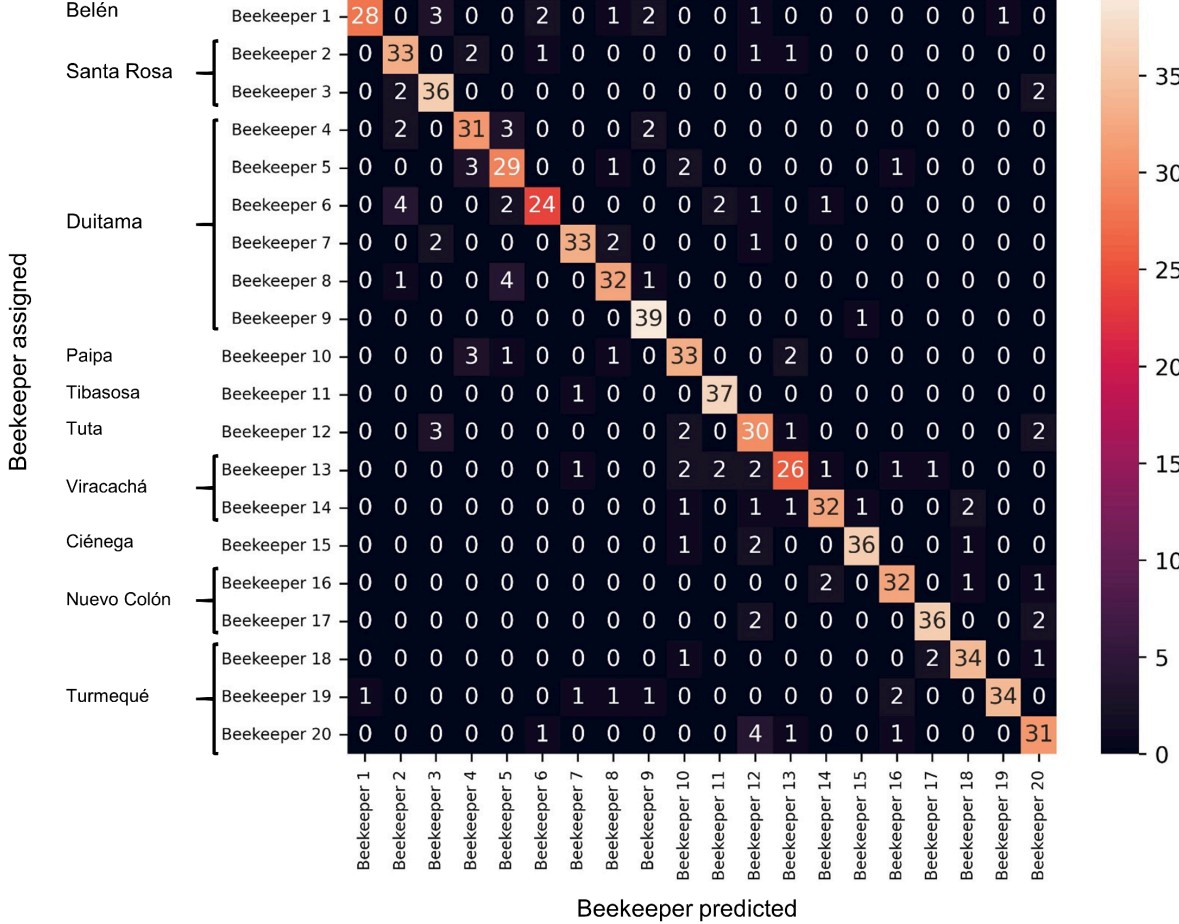

**Fig 8. Confusion matrix for the pollen producer classification on 762 testing pollen images.**

balance between precision and recall when considering all beekeepers. The best results were obtained using an SVM classifier trained with 32 representative color centroids extracted from the *Luv* color space projected to the *uv* dimension. The SVM was configured with a *6th–degree* polynomial kernel, a regularization parameter $C = 0.91$, and an independent coefficient $coef_0 = 1.3$. The SVM consistently produced the best results across evaluation metrics, achieving F1-scores between 0.80 and 0.85. The Random Forest classifier showed moderate performance, with typical F1-scores in the range of 0.70 to 0.78. In contrast, the Multilayer Perceptron exhibited substantial variability and generally underperformed, with F1-scores as low as 0.56 in some configurations. Across all models, increasing the number of centroids tended to improve performance slightly, although the influence of the algorithm type was more significant than the number of centroids.

When the beekeeper association model was employed as a proxy to identify the municipality, i.e., the municipality was determined using the identified beekeeper, the method achieved an accuracy of 88%, a macro-average precision of 86%, a macro-average recall of 85%, and an F1-score of 85%. These results indicate strong performance in the task of identifying the location of the producer.

In Fig 9, we display a scatter plot comparing the F1-score performances for the two test partitions involving the twenty producers considered for the test-retest reliability assessment. The figure illustrates the prediction interval, which represents the range of values likely to contain the value of a new observation under specified predictor settings. As observed, the method exhibited highly similar performances in both test partitions, with narrow confidence intervals. In this case, the Pearson correlation ($r_{1,2}$) was 0.83, indicating good reliability. When assessing the test-retest reliability for municipality location, the Pearson correlation was 0.92, indicating excellent reliability [47,48].

The results, presented in Table 1, show that the model maintained high accuracy, precision, and recall across all periods. These findings suggest that despite potential quantitative shifts in

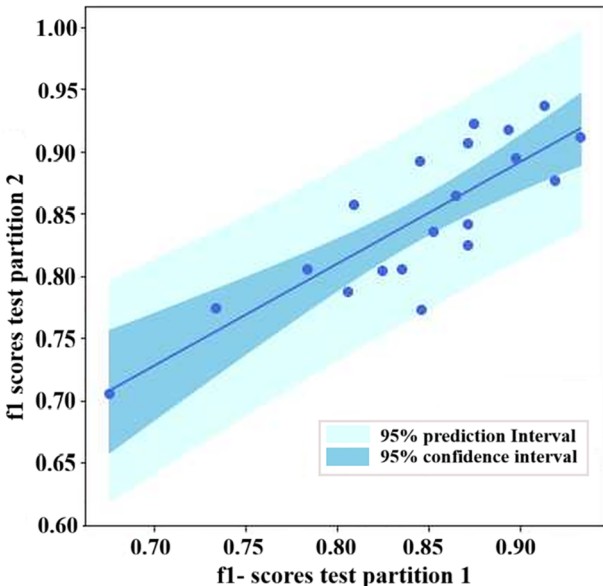

**Fig 9. Test-retest reliability of prediction.** Scatter plot of the performances obtained in each beekeeper for two different predictions used to compute the test-retest reliability.

**Table 1. Performance metrics of the model across temporal sampling visits**

| Visit | Balanced Accuracy | Recall (weighted) | Precision (weighted) | F1 Score (weighted) |
|---|---|---|---|---|
| 1 | 0.88 | 0.88 | 0.88 | 0.87 |
| 2 | 0.84 | 0.84 | 0.85 | 0.84 |
| 3 | 0.78 | 0.78 | 0.80 | 0.78 |
| 4 | 0.75 | 0.74 | 0.75 | 0.73 |

the pollen spectrum, the colorimetric patterns captured by the proposed classification strategy remain stable over time.

## Discussion

### Main contribution and potential

This paper introduces a novel machine learning-based strategy for identifying the geographical origin of corbicular pollen samples using macroscopic product images. The approach relies on a standardized protocol for image sample acquisition, suitable for use by various stakeholders in the pollen production chain. These samples enable the construction of a predictive model that links image information and beekeeper identification by utilizing the similarities in color distributions observed in samples from the same geographical location. In contrast to the traditional approach for characterizing pollen's geographical origin commonly used in palynology, which requires a high level of expertise and expensive equipment, this study provides evidence that more easily obtained pollen color image information at a macroscopic scale can still provide valuable insights into pollen origin.

Due to increased globalization in the food trade, ensuring the integrity of the food chain involves challenges such as origin fraud and quality assurance [49]. Consumers increasingly demand tangible proof of product origins to assess food quality and safety [50]. Associating a product with its origin is fundamental for transparent supply chains, particularly when developing traceability systems [51]. In recent years, the beekeeping industry has experienced rapid growth, leading to an increased demand for traceability systems in supply chains [52–54]. Currently, most traceability-related evidence associated with this type of product comes from specialized labs. In the case of the emerging pollen industry, access to these facilities still needs to be improved, especially in developing countries [55]. Our results show that machine learning strategies can help identify the product's origin and even the specific producer. In particular, identifying the producer resulted in an F1-score of 85%, without relying on palynological or bromatological studies, which are often cost-prohibitive for different supply chain actors. Our results also suggest a high discriminative capacity to distinguish among individual locations, as evidenced by the high individual recall and precision values observed for more than 80% of the studied producers (see Fig 8). Notably, the proposed model also achieved high reliability (see Fig 9), indicating that the proposed association strategy can be applied repeatedly under the same conditions, yielding consistent results, a crucial requirement for product characterization in food systems [56]. These results may contribute to the development of intelligent food traceability systems, which have the potential to significantly improve food safety in global food supply chains [57].

Beyond the specific case of bee pollen, the proposed methodology may be adapted to other domains where product traceability and origin verification are critical. For example, similar image-based approaches could be applied to honey, propolis, or wax, using visual or spectral patterns to identify producers or regions [58]. In agriculture, this technique could support quality control in commodities such as coffee, seeds, or spices, where physical and

visual characteristics often reflect geographic origin and cultivation practices [59]. Applications may also extend to food authentication, particularly in the natural products industry, where origin labeling plays an important role in consumer trust and market value [60].

## Color as a discriminative feature in pollen

This study demonstrates that macroscopic color information provides a reliable basis for classifying bee pollen according to its producer. The proposed model builds upon the well-established role of color in palynological studies [61,62] and extends previous work by demonstrating its applicability to producer-level classification. The observed non-random distribution of pollen colors (see Fig 8) aligns with ecological factors such as plant community composition and pollinator fidelity behavior [63,64], reinforcing the idea that color carries meaningful geographical and botanical information, as observed in the associations in Fig 8). In this case, interestingly, some misclassifications occurred between closely located producers (see Fig 7 for an illustration), suggesting a strong geographical signal in the color patterns acquired at the macroscopic level.

In contrast to earlier efforts that rely on microscopic imaging or physicochemical analyses [16,65–67], our approach focuses on more accessible macroscopic image data, seeking to reduce technical barriers for real-world applications, such as food traceability. By employing a robust and compact color-based representation space, we can effectively capture discriminative features even at a macroscopic-scale acquisition. This relationship is enhanced by considering a controlled acquisition protocol, post-processing to mitigate acquisition artifacts, and a probability-based clustering representation, which proved to be essential for maximizing the quality of the extracted information. The results highlight the potential of color information from macroscopic images as a practical and scalable tool for producer-level pollen classification, opening new possibilities for automated systems in the context of apicultural quality control and traceability based on color.

It should be emphasized that pollen color can vary within a species due to physiological and environmental factors. Our results show that digital images acquired under standardized conditions contain sufficient chromatic information to support origin discrimination at the producer level, see Fig 8 and Table 1. However, the underlying biological mechanisms explaining these patterns are complex and require further investigation. Despite this limitation, our findings provide the first quantitative support for using color as a proxy in pollen origin classification. We recognize that applying this method to other regions or ecosystems may require additional validation. In particular, differences in botanical composition and environmental conditions could affect the relationship between color and origin. Nevertheless, the presence of certain key factors, such as floral fidelity of bees, vegetation stability, well-characterized flowering periods, standardized image pollen collection protocols, and the effectiveness of color-based machine learning algorithms, may facilitate the identification of analogous patterns in new contexts.

Finally, we acknowledge that color represents only one of the potentially discriminative features suitable for describing pollen origin. To enhance the generalizability of classification models, future research could integrate complementary features beyond color. For instance, computational analysis of surface texture may help distinguish visually similar samples, while grain morphology, extracted from macro- or microscopic images, can reveal taxonomically relevant traits. In this context, mobile phone-based microscopy [68] may also offer a low-cost, accessible tool for acquiring such data in field conditions. Additionally, incorporating spectral information from hyperspectral or near-infrared imaging could capture chemical and structural attributes beyond the RGB spectrum, potentially improving classification

performance [69]. However, these advanced techniques may be less feasible for widespread deployment across the production chain, and their integration should consider cost and accessibility constraints.

## Reproducibility

The proposed method shows promising results in classifying bee pollen samples by producer using color information. However, we recognize the importance of evaluating its reproducibility and its potential applicability beyond the specific conditions of this study. The potential for applying this approach to other locations, particularly in tropical regions, is supported by several factors. First, *Apis mellifera* exhibits floral fidelity [63,64], consistently foraging on the same plant species once it has been selected. This behavior may contribute to low intra-annual variability in pollen botanical composition, enhancing consistency in pollen appearance across time and among nearby producers. Second, tropical ecosystems are characterized by continuous or overlapping flowering cycles for most plant species [70], which reduces seasonal variability in pollen sources. These ecological factors support the stability of color-based pollen characteristics, allowing for adequate sampling at regular intervals throughout the year. In addition to these environmental factors, the study implemented a standardized pollen production protocol that producers in other regions can adopt. By maintaining consistency in pollen collection, drying, and image acquisition and processing, variability introduced by processing is minimized, contributing to reproducibility. Additionally, by training on a broad set of reference samples, including those from different flowering periods, the model captures both intra- and inter-taxon color variation. Finally, to evaluate model robustness, we employed both cross-validation and temporal partitioning strategies, testing the model on data from different flowering periods. Considering all these factors in other locations should contribute to the reproducibility of the pattern described here.

## Limitations

The temporal variation in the botanical composition of pollen could potentially affect the association between color and sample identity. However, our results suggest that the colorimetric patterns captured by the model remain stable across different sampling periods, (see Table 1). As previously discussed, this temporal robustness may be linked to the relatively low seasonal turnover in floral sources in the study region and the foraging fidelity of texti-tApis mellifera. Nonetheless, the explicit characterization of the pollen spectrum, particularly its temporal dynamics, could further strengthen the model's interpretability and precision. Incorporating the distribution of pollen types as an intermediate representation may provide additional insights into the underlying biological processes that drive color variation and can enhance the characterization of location. Therefore, we propose this intermediate characterization of pollen as a direction for future work.

The study was conducted in a single geographic region within the department of Boyacá, Colombia (see Fig 7). Logistical and budgetary constraints primarily drove the decision to focus on this area. However, the proposed methodology was designed to be reproducible and scalable, with standardized protocols at each stage of data collection and analysis. It is worth noting that while the classification model in this study was trained and tested using data from producers in a single region, the conditions under which the method was developed, floral fidelity, and low temporal pollen variability are not exclusive to the area under study and are commonly found across other tropical beekeeping zones [71,72]. However, we recognize the importance of evaluating the generalizability of this approach beyond the current context. Therefore, future work will involve collaboration with producer associations in other tropical

regions to assess the model's performance in new environments, validate its robustness, and explore the integration of additional features such as texture or spectral information to improve classification in more diverse ecological settings.

It is worth noting that many studies in the fields of palynology, food traceability, and spectroscopic classification have focused on single geographic regions when developing and validating new methodologies. This approach is common in early-stage or feasibility studies, where logistical and ecological consistency is essential to isolate key variables and establish robust protocols. For example,in [73] bee pollen from a single region in west-central Poland was characterized using ED-XRF and ATR-FTIR spectroscopy, achieving consistent elemental and organic profiles across samples. Similarly, Fourier Transform Infrared Spectroscopy and Raman spectroscopy have been used effectively to classify pollen types from a single location based on vibrational signatures [74]. In [75], FTIR spectroscopy was applied to 126 bee pollen samples from a single Portuguese region to assess compositional traits and verify botanical origin. These studies demonstrate that scientifically rigorous and reproducible classification systems can be developed within localized contexts.

It is important to note that the classifier developed in this study was trained on a fixed set of producers. Therefore, when a new producer is to be included for the characterization of its location, the proposed model should be retrained to incorporate samples from that producer into the classification framework [76]. While this requirement does imply an additional step for extending the system, it does not undermine the validity or utility of the approach presented here. We recall that our primary contribution lies in demonstrating that a relatively simple, accessible, and reproducible image-based method can effectively classify corbicular pollen samples by origin under standardized conditions. The results serve as proof of concept that color-based features extracted from macroscopic images hold strong discriminatory power for pollen origin. Looking forward, this approach may be adapted to classify samples at broader geographical scales, such as ecological zones or regional denominations of origin, rather than individual producers, which could enhance its scalability and reduce the need for frequent retraining.

This study demonstrates that quantitative color analysis can complement traditional morphological features by reducing the subjectivity associated with human color perception [77], for instance, in the perception of similar colors, even when they originate from different plants. Pollen grains were imaged under controlled illumination and converted into informative color spaces, enabling the classifier to capture subtle chromatic features associated with botanical origin. The variety of training samples also helps capture subtle color variations. However, in some cases, color alone may not suffice to distinguish between taxa. Future work may include confidence estimation frameworks, such as conformal prediction [78], to enhance the interpretability and reliability of classification outcomes.

## Future directions

From the methodological perspective, more recent approaches, such as those based on deep learning, can also be studied to address the problem of color characterization [79]. Most of these methods rely on automatically constructing the representation space by optimizing the model to adjust to large amounts of data. Therefore, related approaches based on transfer learning can also be explored [79]. However, these alternatives require greater interpretability and may increase the risk of overfitting compared to less complex machine learning classification approaches [79]. Future work may also focus on expanding the dataset, exploring more advanced deep learning techniques, and investigating the potential of transfer learning in this context.

Another area to explore is the characterization of pollen samples through DNA barcoding analysis. Studies have demonstrated that this technique is a powerful tool for determining both the geographical and botanical origin of honey samples, relying on the analysis of pollen residues [80,81]. However, one limitation of this method should be noted: despite its effectiveness, the associated costs are not easily affordable [82].

The quality of the results can be highly dependent on the image acquisition conditions. Even though the proposed protocol can be adapted for the traceability task, future work may consider acquiring images under in less controlled conditions. In this case, deep learning-based methods can potentially be used to address the characterization problem [83]. Future work may also explore the model's generalization ability across different acquisition devices. Forthcoming research also involves the use of pollen spectrum studies to associate pollen types and their quantities in each sample, with the aim of improving prediction accuracy.

## Author contributions

**Conceptualization:** Carlos A. Martínez Niño, Francisco Gómez Jaramillo.

**Data curation:** Juan D. Leal Campuzano.

**Formal analysis:** Juan D. Leal Campuzano, Carlos A. Martínez Niño.

**Funding acquisition:** Carlos A. Martínez Niño, Francisco Gómez Jaramillo.

**Investigation:** Juan D. Leal Campuzano, Francisco Gómez Jaramillo.

**Methodology:** Juan D. Leal Campuzano, Carlos A. Martínez Niño, Francisco Gómez Jaramillo.

**Resources:** Francisco Gómez Jaramillo.

**Software:** Juan D. Leal Campuzano.

**Supervision:** Carlos A. Martínez Niño, Francisco Gómez Jaramillo.

**Visualization:** Juan D. Leal Campuzano.

**Writing – original draft:** Juan D. Leal Campuzano, Carlos A. Martínez Niño, Francisco Gómez Jaramillo.

**Writing – review & editing:** Juan D. Leal Campuzano, Carlos A. Martínez Niño, Francisco Gómez Jaramillo.

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
