## [Decision Letter · Decision Letter 0]

12 Mar 2025

PONE-D-24-51855Classification of images of bee pollen according to their producersPLOS ONE

Dear Dr. Gómez,

Thank you for submitting your manuscript to PLOS ONE. After careful consideration, we feel that it has merit but does not fully meet PLOS ONE’s publication criteria as it currently stands. Therefore, we invite you to submit a revised version of the manuscript that addresses the points raised during the review process.

We look forward to receiving your revised manuscript.

Kind regards,

Kai Wang

Academic Editor

PLOS ONE

**Journal Requirements:**

Please ensure that your manuscript meets PLOS ONE's style requirements, including those for file naming. The PLOS ONE style templates can be found at https://journals.plos.org/plosone/s/file?id=wjVg/PLOSOne_formatting_sample_main_body.pdf and https://journals.plos.org/plosone/s/file?id=ba62/PLOSOne_formatting_sample_title_authors_affiliations.pdf 2. Please update your submission to use the PLOS LaTeX template. The template and more information on our requirements for LaTeX submissions can be found at http://journals.plos.org/plosone/s/latex. 3. Please note that PLOS ONE has specific guidelines on code sharing for submissions in which author-generated code underpins the findings in the manuscript. In these cases, we expect all author-generated code to be made available without restrictions upon publication of the work. Please review our guidelines at https://journals.plos.org/plosone/s/materials-and-software-sharing#loc-sharing-code and ensure that your code is shared in a way that follows best practice and facilitates reproducibility and reuse. 4. Thank you for stating the following in your Competing Interests section:  “NO authors have competing interests” Please complete your Competing Interests on the online submission form to state any Competing Interests. If you have no competing interests, please state "The authors have declared that no competing interests exist.", as detailed online in our guide for authors at http://journals.plos.org/plosone/s/submit-nowThis information should be included in your cover letter; we will change the online submission form on your behalf. 5. In the online submission form, you indicated that The data used in this study are available from the authors upon reasonable request. All PLOS journals now require all data underlying the findings described in their manuscript to be freely available to other researchers, either a. In a public repository, b. Within the manuscript itself, or c. Uploaded as supplementary information.This policy applies to all data except where public deposition would breach compliance with the protocol approved by your research ethics board. If your data cannot be made publicly available for ethical or legal reasons (e.g., public availability would compromise patient privacy), please explain your reasons on resubmission and your exemption request will be escalated for approval. 6. When completing the data availability statement of the submission form, you indicated that you will make your data available on acceptance. We strongly recommend all authors decide on a data sharing plan before acceptance, as the process can be lengthy and hold up publication timelines. Please note that, though access restrictions are acceptable now, your entire data will need to be made freely accessible if your manuscript is accepted for publication. This policy applies to all data except where public deposition would breach compliance with the protocol approved by your research ethics board. If you are unable to adhere to our open data policy, please kindly revise your statement to explain your reasoning and we will seek the editor's input on an exemption. Please be assured that, once you have provided your new statement, the assessment of your exemption will not hold up the peer review process. 7. We note that Figures 1 and 6 in your submission contain map images which may be copyrighted. All PLOS content is published under the Creative Commons Attribution License (CC BY 4.0), which means that the manuscript, images, and Supporting Information files will be freely available online, and any third party is permitted to access, download, copy, distribute, and use these materials in any way, even commercially, with proper attribution. For these reasons, we cannot publish previously copyrighted maps or satellite images created using proprietary data, such as Google software (Google Maps, Street View, and Earth). For more information, see our copyright guidelines: http://journals.plos.org/plosone/s/licenses-and-copyright. We require you to either present written permission from the copyright holder to publish these figures specifically under the CC BY 4.0 license, or remove the figures from your submission: a. You may seek permission from the original copyright holder of Figures 1 and 6 to publish the content specifically under the CC BY 4.0 license.   We recommend that you contact the original copyright holder with the Content Permission Form (http://journals.plos.org/plosone/s/file?id=7c09/content-permission-form.pdf) and the following text:“I request permission for the open-access journal PLOS ONE to publish XXX under the Creative Commons Attribution License (CCAL) CC BY 4.0 (http://creativecommons.org/licenses/by/4.0/). Please be aware that this license allows unrestricted use and distribution, even commercially, by third parties. Please reply and provide explicit written permission to publish XXX under a CC BY license and complete the attached form.” Please upload the completed Content Permission Form or other proof of granted permissions as an "Other" file with your submission. In the figure caption of the copyrighted figure, please include the following text: “Reprinted from [ref] under a CC BY license, with permission from [name of publisher], original copyright [original copyright year].” b. If you are unable to obtain permission from the original copyright holder to publish these figures under the CC BY 4.0 license or if the copyright holder’s requirements are incompatible with the CC BY 4.0 license, please either i) remove the figure or ii) supply a replacement figure that complies with the CC BY 4.0 license. Please check copyright information on all replacement figures and update the figure caption with source information. If applicable, please specify in the figure caption text when a figure is similar but not identical to the original image and is therefore for illustrative purposes only.The following resources for replacing copyrighted map figures may be helpful: USGS National Map Viewer (public domain): http://viewer.nationalmap.gov/viewer/The Gateway to Astronaut Photography of Earth (public domain): http://eol.jsc.nasa.gov/sseop/clickmap/Maps at the CIA (public domain): https://www.cia.gov/library/publications/the-world-factbook/index.html and https://www.cia.gov/library/publications/cia-maps-publications/index.htmlNASA Earth Observatory (public domain): http://earthobservatory.nasa.gov/Landsat: http://landsat.visibleearth.nasa.gov/USGS EROS (Earth Resources Observatory and Science (EROS) Center) (public domain): http://eros.usgs.gov/#Natural Earth (public domain): http://www.naturalearthdata.com/

**Additional Editor Comments:**

Please revise your manuscript following the reviewers' comments. Please note that Reviewer 2 recommend the rejection on your paper, so I suggest you need to respond his suggestions in a careful manner.

Reviewers' comments:

Reviewer's Responses to Questions

**Comments to the Author**

1. Is the manuscript technically sound, and do the data support the conclusions?

Reviewer #1: Yes

Reviewer #2: Partly

Reviewer #3: Yes

2. Has the statistical analysis been performed appropriately and rigorously? 

Reviewer #1: No

Reviewer #2: Yes

Reviewer #3: Yes

3. Have the authors made all data underlying the findings in their manuscript fully available?

Reviewer #1: Yes

Reviewer #2: Yes

Reviewer #3: No

4. Is the manuscript presented in an intelligible fashion and written in standard English?

Reviewer #1: Yes

Reviewer #2: Yes

Reviewer #3: Yes

5. Review Comments to the Author

**Reviewer #1: **General Comments

1/I am not convince how the technique is consistent/reproducibility and can be applied in other areas. Apart from color differentiation any other parametes can be considered as well?

2/Can author show at least two different location to show elements of reproducibility

Specific commments: per location how long you can get results?

**Reviewer #2: **Please, see the file with comments.

Remember to consider some aspects of pollen biology and bee behavior that may influence the coloration of the collected pollen. Additionally, there are external factors, such as the drying method, which can also interfere with the coloration.

**Reviewer #3: **The manuscript presents a novel computer vision approach for the identification of bee pollen origin based solely on images of pollen grains. The method is well defined and the manuscript is well written. Both the acquisition hardware and the image analysis are described, making it applicable in a real setting. But there are some corrections the authors need to do to make the paper more solid:

1. The method is based on the hypothesis that each producer / region has a distinct color trait. The authors should explain in more detail the cause of these color patterns. As stated in page 7, “a relationship exists between the color of the pollen and the species from which it originates”. Is pollen color constant for a certain species?

2. Following on question 1, does the approach assume that the combination of plan species is unique for every producer or region? Is this always the case?

3. Section 2, Sample Image Acquisition: The authors write that “the grains were evenly spaced”. How is this spreading performed? How could it affect the acquired images and the classification result?

4. Section 2, Sample Image Acquisition: The authors should give more details on the camera used in the experiments. At the end of the Discussion section, they mention the generalization to other devices as future research, but they don’t give details on the camera or the minimum technical requirements needed to replicate their experiments.

5. Section 2.5., Color feature extraction: The number of centroids for each producer is determined by the number of samples detected in a palynological analysis. Is this number maintained constant for each producer?

6. Section 2.6: The authors indicate that “various multiclass classification strategies were explored” that included support vector machines and perceptrons. But the method used to obtain the final results presented in Section 3 is not specified. For the sake of reproducibility, the authors should:

a. Indicate what was the actual method that gave the results presented in the paper

b. The input vector has a different size for each region, depending on the number of species detected. How is this

difference handled in the machine learning algorithm?

c. Give details on the implementation of the method (parameters that need to be tuned, etc.). Only a a final best result in provided in the paper. What was the variability in the results with different machine learning techniques?

7. Section 2.7, Experimental evaluation. The authors should describe the distribution of producer among the final set of samples used in the experiments.

8. From a translational point of view: If a new producer enters the market, the system needs to be retrained to include the new producer. The authors should comment on this in the Discussion section.

6. PLOS authors have the option to publish the peer review history of their article (what does this mean?). If published, this will include your full peer review and any attached files.

Reviewer #1: **Yes: **Fahrul Huyop

Reviewer #2: No

Reviewer #3: No

---

## [Author Response · Author response to Decision Letter 1]

13 Aug 2025

Reviewer # 1

1./I am not convinced how the technique is consistent/reproducibility and can be applied in other areas. Apart from color differentiation, can any other parameters be considered as well? Response: We thank the reviewer for raising the important issue of consistency, reproducibility, and general applicability of our method. We want to remark that for ensuring consistency and reproducibility, we employed a standardized and documented protocol for pollen collection, image acquisition, and storage conditions, minimizing variability due to environmental or handling factors. Additionally, the image analysis pipeline was fully automated, and the model was trained and validated using cross-validation and temporal partitioning strategies. These measures allowed us to assess the model’s robustness. For clarifying these points, the last version of the manuscript provides a highly detailed description of all steps required to reproduce the proposed approach. In particular: the second section outlines the procedures required to reproduce the proposed model, sub-section 2.1 describes the experimental design used for data collection, along with contextual information regarding the origin of the data, in sub-section 2.2, the tools developed for the study are introduced, as well as the standardized protocol applied to both the acquisition of representative data and the photographic capture process, sub-section 2.3 provides an overview of the preprocessing methods employed, sub-section 2.4 presents the theoretical foundations underlying the use of color as a means of data representation. The subsequent sections propose various classification models, which can be adapted in future researchers aiming to reproduce this process.

In addition, in order to guarantee computational reproducibility of the results we also included all the computer code and the complete dataset required for reproducing the reported results. This information reads as follows in the last version of the manuscript in the Materials and Methods section (lines 303):

“The used code and the image database needed to reproduce results can be accessed in https://github.com/JuanDavid1703/PolenProject.git.”

Related with the application to other areas, we acknowledge that generalization to other geographic regions or botanical contexts may require additional adaptations. In regions with greater floral diversity or different beekeeping practices, the relationship between color and pollen origin may vary. In addition, we included a novel subsection discussion the main points related to reproducibility in Discussion section as follows (lines 434 - 454):

“4.3 Reproducibility The proposed method shows promising results in classifying bee pollen samples by producer using color information. However, we recognize the importance of evaluating its reproducibility and its potential applicability beyond the specific conditions of this study. The potential for applying this approach to other locations, particularly in tropical regions, is supported by several factors. First, Apis mellifera exhibits floral fidelity (Lau and Galloway, 2004; Jorgensen et al, 2006), consistently foraging on the same plant species once it has been selected. This behavior may contribute to low intra-annual variability in pollen botanical composition, a feature that enhances consistency in pollen appearance across time and among nearby producers. Second, tropical ecosystems are characterized by continuous or overlapping flowering cycles for most plant species (Instituto de Investigación de Recursos Biologicos Alexander von Humboldt, 2020), which reduces seasonal variability in pollen sources. These ecological factors support the stability of color-based pollen characteristics, allowing for adequate sampling at regular intervals throughout the year. In addition to these environmental factors, the study implemented a standardized pollen production protocol that producers in other regions can adopt. By maintaining consistency in pollen collection, drying, and imaging acquisition and processing, variability introduced by processing is minimized, contributing to reproducibility. Additionally, by training on a broad set of reference samples, including those from different flowering periods, the model captures both intra- and inter-taxon color variation. Finally, to evaluate model robustness, we employed both cross-validation and temporal partitioning strategies, testing the model on data from different flowering periods. Considering all these factors in other locations should contribute to the reproducibility of the pattern described here.”

To enhance generalizability and reduce reliance on color alone, future work may incorporate complementary features, including, texture descriptors (e.g., local binary patterns or Gabor filters) to capture surface characteristics, morphological features derived from microscopic or macroscopic grain analysis, and spectral data obtained through hyperspectral or near-infrared imaging to capture chemical and structural variations not visible in RGB images. For accounting for this relevant point we include in the Discussion section the following text (lines 423-433):

“Finally, we acknowledge that color represents only one of the potentially discriminative features suitable to describe pollen origin. To enhance the generalizability of classification models, future research could integrate complementary features beyond color. For instance, computational analysis of surface texture may help distinguish visually similar samples, while grain morphology, extracted from macro- or microscopic images, can reveal taxonomically relevant traits. In this case, mobile phone-based microscopy (Switz et al, 2014) may also offer again a low-cost, accessible tool for acquiring such data in field conditions. Additionally, incorporating spectral information from hyperspectral or near-infrared imaging could capture chemical and structural attributes beyond the RGB spectrum, potentially improving classification performance (Sendin et al, 2018). However, these advanced techniques may be less feasible for widespread deployment across the production chain, and their integration should consider cost and accessibility constraints.”

In addition, beyond bee pollen characterization, the approach has potential for adaptation to other domains. For example, provenance verification in honey, propolis, or wax by analyzing visual or spectral traits associated with origin, quality control in agricultural commodities (e.g., coffee, spices, or seeds), where visual or physical characteristics are critical indicators of geographic or botanical origin, traceability tools for food authentication, particularly in natural products where certification of origin impacts market value. We thank the referee for calling our attention to these highly relevant potential areas of application, which are now considered in the Discussion section (381-388).

“Beyond the specific case of bee pollen, the proposed methodology may be adapted to other domains where product traceability and origin verification are critical. For example, similar image-based approaches could be applied to honey, propolis, or wax, using visual or spectral patterns to identify producers or regions (Shafiee et al, 2014). In agriculture, this technique could support quality control in commodities such as coffee, seeds, or spices, where physical and visual characteristics often reflect geographic origin and cultivation practices (Zhu et al, 2021). Applications may also extend to food authentication, particularly in the natural products industry, where origin labeling plays an important role in consumer trust and market value (Wu et al, 2021).”

2/Can the author show at least two different locations to show elements of reproducibility? Response: We thank the reviewer for the suggestion to include results from more than one location to assess reproducibility. Due to logistical and budgetary constraints, this study was limited to a single geographic region within the department of Boyacá (Colombia). We agree that extending the method to additional locations would strengthen generalizability and reproducibility. However, our approach aligns with well-established practices in the field, where initial validation is often performed in a single location to control variability and demonstrate feasibility. Studies such as Malinowski et al. (2021), Zheleznyak et al., 2021, and Estevinho et al. (2017) similarly developed and tested characterization methods using samples from a single site. These works are frequently cited and serve as precedent for the current study’s design. We have added a paragraph in the Discussion section to acknowledge this context and clarify how future work will address multi-site validation.

“The study was conducted in a single geographic region within the department of Boyacá, Colombia (see Figure 7). Logistical and budgetary constraints primarily drove the decision to focus on this area. However, the proposed methodology was designed to be reproducible and scalable, with standardized protocols at each stage of data collection and analysis. It is worth recalling that while the classification model in this study was trained and tested using data from producers in a single region, the conditions under which the method was developed, floral fidelity, and low temporal pollen variability are not exclusive to the area under study and are commonly found across other tropical beekeeping zones (Amaya, 2009; Amakpe et al, 2024). However, we recognize the importance of evaluating the generalizability of this approach beyond the current context. Therefore, future work will involve collaboration with producer associations in other tropical regions to assess the model's performance in new environments, validate its robustness, and explore the integration of additional features such as texture or spectral information to improve classification in more diverse ecological settings. It is worth noting that many studies in the fields of palynology, food traceability, and spectroscopic classification have focused on single geographic regions when developing and validating new methodologies. This approach is prevalent in early-stage or feasibility studies, where logistical and ecological consistency is essential to isolate key variables and establish robust protocols. For example, Swiatly-Blaszkiewicz et al (2021) characterized bee pollen from a single region in west-central Poland using ED-XRF and ATR-FTIR spectroscopy, achieving consistent elemental and organic profiles across samples. Similarly, Fourier Transform Infrared Spectroscopy and Raman spectroscopy have been used effectively to classify pollen types from a single location based on vibrational signatures (Kendel and Zimmermann, 2020). Anjos et al (2017) also applied Fourier Transform Infrared - Attenuated Total Reflectance spectroscopy to 126 bee pollen samples from a single Portuguese region to assess compositional traits and verify botanical origin. These studies demonstrate that scientifically rigorous and reproducible classification systems can be developed within localized contexts.”

Finally, while our results are specific to a set of producers in one region, the methodology was designed with controlled, standardized, and reproducible components, making it adaptable to other tropical regions with similar ecological and production characteristics. Specifically, our approach relies on (1) the floral fidelity of Apis mellifera, which results in low inter-annual variability in pollen composition; (2) the continuous flowering cycles typical of tropical ecosystems, which allow for consistent pollen collection throughout the year; and (3) a standardized protocol for pollen collection, drying, imaging, and processing. These conditions are not unique to the Boyacá region and can be reasonably replicated by producers across other tropical areas, as discussed in section Reproducibility in the last version of the manuscript.

Reviewer #2

Remember to consider some aspects of pollen biology and bee behavior that may influence the coloration of the collected pollen. Additionally, there are external factors, such as the drying method, which can also interfere with the coloration. Response: We highly appreciate your valuable observations about the importance of considering pollen biology, bee behaviour and pollen preparation as possible sources of changes of color. We agree with the referee about the importance of considering these elements. In our study, we focused on Apis mellifera, a species well known for its floral constancy during foraging. According to Heuel et al (2024); Gonzalves et al (2018a); Sanderson and Wells (2005), once a foraging bee selects a particular floral source, it tends to continue collecting pollen from the same species over extended periods. This behavior leads to reduced intra-colony variability in pollen origin and, consequently, a high degree of botanical uniformity in pollen samples. While we acknowledge that minor biochemical transformations can occur during pollen collection and packing by bees, the dominant influence on pollen color remains botanical origin, in fact, this is the main observation underlying the colorimetric work commonly devised in the palynology domain (Erdman 1969, Faegri 1989, Moore et al. 1991). To address this point, we included the following statement in the manuscript in the Materials and Methods section:

“It is worthy recall that honeybee foraging behavior, particularly in Apis mellifera, is characterized by floral constancy, whereby worker bees consistently visit the same plant species once selected. This behavior, supported by Heuel et al (2024); Gonzalves et al (2018a); Sanderson and Wells (2005), results in minimal variability in the botanical origin of collected pollen over time, thus reducing color variation attributable to plant species,particularly those that provide the most nutrients for the hive (Waser, 1986).”

In addition, we acknowledge that the pollen nature is an essential determinant of the color information herein characterized. Therefore, we also included a complete description of the rationale underlying the pollen sample collection, emphasizing the primary role of color stability of pollen in tropical regions. These changes read as follows in the Materials and Methods section (lines 115-130):

“This study involved beekeepers from various locations in the Márquez region, including the municipalities of Nuevo Colón, Turmequé, Viracachá, and Ciénega, and the Tundama region, encompassing the municipalities of Tibasosa, Tuta, Belén, Santa Rosa de Viterbo, Duitama, Paipa, and Tuta, all located within the Boyacá department of Colombia. These municipalities are located in regions known for high pollen production. For the sample acquisition, a comprehensive survey was first conducted to determine the flowering periods of the plant species present in the ecosystems relevant to the project. This information was essential for subsequently identifying the origin of the pollen collected by the colonies in the participating apiaries.A one-year collection calendar was developed in collaboration with local beekeepers and two local experts in bee production systems with field experience comprising four distinct pollen sampling periods. To set this number of samples, the producers and two local experts in bee production systems with field experience considered the variations in the flowering cycle of the region under study. It is worth mentioning that in tropical areas as the region under study, in contrast to other regions, there are no marked seasons, and consequently, most plants visited by bees blossom the whole (Araújo et al. 2022). This factor contributes to reducing the floristic variability along the year, helping to minimize the number of visits for collecting samples.

We agree that the drying process can potentially affect pollen color if not properly standardized. For this reason, all participating pollen producers followed a unified protocol based on Vásquez et al. (2021), which includes precise drying parameters: stainless steel dryers, temperatures between 50–60 °C, and durations from 12 to 24 hours. For this study, all samples were dried for 24 hours at 55 °C, resultin

---

## [Decision Letter · Decision Letter 1]

27 Aug 2025

PONE-D-24-51855R1Classification of images of bee pollen according to their producersPLOS ONE

Dear Dr. Gómez,

Thank you for submitting your manuscript to PLOS ONE. After careful consideration, we feel that it has merit but does not fully meet PLOS ONE’s publication criteria as it currently stands. Therefore, we invite you to submit a revised version of the manuscript that addresses the points raised during the review process.

We look forward to receiving your revised manuscript.

Kind regards,

Kai Wang

Academic Editor

PLOS ONE

Journal Requirements:

Additional Editor Comments:

Please note that the first reivewer who recommend minor suggestion suggest the rejection this time. You might respond to his concerns.

Language editing also is suggested by the second reviewer. Please addess this point.

Kai Wang, editor. 

Reviewers' comments:

Reviewer's Responses to Questions

**Comments to the Author**

1. If the authors have adequately addressed your comments raised in a previous round of review and you feel that this manuscript is now acceptable for publication, you may indicate that here to bypass the “Comments to the Author” section, enter your conflict of interest statement in the “Confidential to Editor” section, and submit your "Accept" recommendation.

Reviewer #1: All comments have been addressed

Reviewer #2: All comments have been addressed

2. Is the manuscript technically sound, and do the data support the conclusions?

Reviewer #1: No

Reviewer #2: Yes

3. Has the statistical analysis been performed appropriately and rigorously? 

Reviewer #1: I Don't Know

Reviewer #2: Yes

4. Have the authors made all data underlying the findings in their manuscript fully available?

Reviewer #1: No

Reviewer #2: Yes

5. Is the manuscript presented in an intelligible fashion and written in standard English?

Reviewer #1: Yes

Reviewer #2: Yes

6. Review Comments to the Author

Reviewer #1: General Comments:

1.There is always limitations of color-based identification on pollen.

2.Therefore, identification based solely on pollen color is limited, as color can vary within a single species and may also overlap across different species. As such, it does not provide a precise or reliable basis for pollen classification.

3. More advanced methods, such as pollen DNA metabarcoding, have been widely demonstrated by many researchers to be effective in tracing both the geographical and botanical origins of pollen.

4. Generally, Bee pollen is widely recognized for its health benefits, and its nutritional and bioactive properties varying by botanical origin.

Reviewer #2: I recommend accepting the manuscript after language editing by a native speaker to improve some passages that may be poorly placed, but this consideration is also from someone who is not a native English speaker.

7. PLOS authors have the option to publish the peer review history of their article (what does this mean?). If published, this will include your full peer review and any attached files.

Reviewer #1: No

Reviewer #2: No

---

## [Author Response · Author response to Decision Letter 2]

26 Sep 2025

PONE-D-24-51855R1

Classification of images of bee pollen according to their producers

Plos One

Dear Plos One Editor,

We thank the reviewers for all the thoughtful comments and suggestions that helped improve our work's presentation and clarity. In particular, we appreciate the very careful reading of our manuscript. We thank you for the comments and hope our modifications and responses fulfill the requirements. For ease of review, all our responses are marked in green, and the corresponding modifications in the manuscript appear in blue.

Reviewers' comments and responses:

Reviewer #1: General Comments:

1.There are always limitations of color-based identification on pollen.

We thank the referee for this highly valuable comment. We acknowledge that while color may present some limitations when establishing a digital fingerprint of the botanical origin of a pollen sample, the findings of this study demonstrate that, using color information acquired on standardized conditions, it is possible to predict its geographical origin based on color. To address this problem, two local experts in bee production worked with the producers to develop a “floral calendar” that documents flowering dynamics throughout the year. Moreover, because this is a tropical region, most plant species flower continuously, which reduces seasonal variability. These factors support the validity of our findings and provide a baseline for future research.

As discussed in the manuscript, for the present study, the available economic and logistical resources enabled analysis within a one-year range. A natural extension of this methodology would involve broadening the temporal scope and incorporating additional variables, such as nutritional composition or laboratory-derived features, while maintaining a favorable cost–benefit ratio to ensure feasibility for beekeepers with limited resources. Such work would allow us to verify whether significant color variation arises in relation to botanical origin beyond the conditions studied here.

2.Therefore, identification based solely on pollen color is limited, as color can vary within a single species and may also overlap across different species. As such, it does not provide a precise or reliable basis for pollen classification.

We appreciate this important observation. Pollen color can indeed vary within a species and overlap across different species, which limits its precision for species-level identification. However, it is important to clarify that the main objective of our study was not to identify the botanical species present in the samples, but rather to classify the producer. In this context, our results indicate that the proposed methodology achieves a reasonable level of accuracy for this task.

We fully recognize the inherent limitations of relying solely on color information, and this was precisely part of the research question: given that this is a cost-effective and accessible approach, how well can it perform in predicting the sample origin at the producer level? The results obtained so far suggest that it provides an appropriate predictive performance for this specific classification problem. Future work will naturally extend this framework to incorporate complementary features for species-level identification.

3. More advanced methods, such as pollen DNA metabarcoding, have been widely demonstrated by many researchers to be effective in tracing both the geographical and botanical origins of pollen.

We appreciate your suggestion regarding the use of DNA metabarcoding, as it is indeed a powerful and widely recognized method for tracing both the botanical and geographical origin of pollen. Nevertheless, one of the aims of our work is to propose a cost–time efficient alternative. In Colombia, and particularly within the beekeeping sector and pollen production chain, there is currently limited economic capacity to implement DNA-based methodologies or other resource-intensive analytical processes.

Recent advances in machine learning and computer vision have shown that color, when captured and analyzed with appropriate models, can serve as a reliable digital fingerprint. This work contributes to this literature. Although imperceptible to the human eye, subtle chromatic patterns can be effectively detected through accessible tools such as smartphones or cameras, enabling accurate predictions at a fraction of the cost. For example, whereas DNA metabarcoding services typically range between USD 100–250 per sample (University of North Texas, 2024–2025), the approach presented here operates at less than USD 1 per sample after model deployment, highlighting its practical advantages for producers with limited resources.

It is also important to note that DNA metabarcoding is not yet commonly available in Colombia and in many tropical regions. Beyond cost, there are technical barriers: many primers used in global metabarcoding protocols are not optimized for the high biodiversity of Colombian flora, and numerous local species remain underrepresented in existing reference libraries. These constraints limit the immediate applicability of DNA barcoding in our context, as confirmed by local experts in bee science.

Nonetheless, we fully recognize the value of DNA-based methods as a complementary tool. In the revised manuscript, we explicitly acknowledge DNA metabarcoding as a promising direction for future work, while emphasizing that our current proposal offers a cost-effective, accessible, and scalable alternative that can already support traceability in developing regions..

The following comment will be added to the document:

Another area to explore is the characterization of pollen samples through DNA barcoding analysis. Studies have demonstrated that this technique is a powerful tool for determining both the geographical and botanical origin of honey samples, relying on the analysis of pollen residues (Ruppert, et al, 2019; Ullah and Huyop, 2024) . However, one limitation of this method should be noted: despite its effectiveness, the associated costs are not easily affordable (University of North Texas, 2024–2025).

4. Generally, Bee pollen is widely recognized for its health benefits, and its nutritional and bioactive properties varying by botanical origin.

We thank the referee for this important observation. In our study, we acknowledged that pollen is a key resource for sustaining bee colonies and that it also provides essential nutritional properties beneficial to humans (Oliveira et al., 2009). A promising research direction would be to investigate whether the botanical origin and nutritional composition of a sample can be inferred from its color characteristics, given that compounds such as isoflavonoids and carotenoids are simultaneously linked to pollen pigmentation and nutritional value (Velásquez et al., 2025; Rzepecka-Stojko et al., 2015). In this regard, the potential of color analysis to serve as a proxy for botanical origin and associated nutritional traits represents an attractive avenue for future work (Gámbaro et al., 2025).

Reviewer #2 General comments:

1. I recommend accepting the manuscript after language editing by a native speaker to improve some passages that may be poorly placed, but this consideration is also from someone who is not a native English speaker.

Thank you for your suggestions. The document has been shared with a native scientific language editor for proper review and style correction. The revisions were highlighted in blue, consistent with the approach previously applied to the paragraphs appended to the article. Thanks for your observations.

References:

Simão C. et al. (2025). Digital image processing + ML as an effective approach for bee pollen classification.

University of North Texas. (2025). Genomics Center: Pricing. UNT Research. https://research.unt.edu/resources/research-core-facilities/genomics-center/pricing.html.

Velásquez, P., Muñoz-Carvajal, E., Grimau, L., Bustos, D., Montenegro, G., & Giordano, A. (2023). Floral Pollen Bioactive Properties and Their Synergy in Honeybee Pollen. Chemistry & biodiversity, 20(4), e202201138. https://doi.org/10.1002/cbdv.202201138

Rzepecka-Stojko, A., Stojko, J., Kurek-Górecka, A., Górecki, M., Kabała-Dzik, A., Kubina, R., Moździerz, A., & Buszman, E. (2015). Polyphenols from Bee Pollen: Structure, Absorption, Metabolism and Biological Activity. Molecules, 20(12), 21732-21749. https://doi.org/10.3390/molecules201219800

Oliveira, K. C. L. S., Moriya, M., Azedo, R. A. B., Almeida-Muradian, L. B., Teixeira, E. W., Alves, M. L. T. M. F., & Moreti, A. C. C. (2009). Relationship between botanical origin and antioxidants vitamins of bee-collected pollen. Química Nova, 32(5), 1099-1102. https://doi.org/10.1590/S0100-40422009000500003

Gámbaro, A., Miraballes, M., Urruzola, N., Kniazev, M., Dauber, C., Romero, M., Fernández-Fernández, A. M., Medrano, A., Santos, E., & Vieitez, I. (2025). Physicochemical Composition and Bioactive Properties of Uruguayan Bee Pollen from Different Botanical Sources. Foods, 14(10), 1689. https://doi.org/10.3390/foods14101689

Ruppert, K. M., Kline, R. J., & Rahman, M. S. (2019). Past, present, and future perspectives of environmental DNA (eDNA) metabarcoding: A systematic review in methods, monitoring, and applications of global eDNA. Global Ecology and Conservation, 17, e00547. https://doi.org/10.1016/j.gecco.2019.e00547

Ullah, S., & Huyop, F. (2024). Using pollen DNA metabarcoding to trace geographical origin of honey; all identified taxa were indigenous to the region. Meta Gene, 38, 101012. https://doi.org/10.1016/j.mgene.2024.101012

---

## [Decision Letter · Decision Letter 2]

30 Sep 2025

Classification of images of bee pollen according to their producers

PONE-D-24-51855R2

Dear Dr. Gómez,

We’re pleased to inform you that your manuscript has been judged scientifically suitable for publication and will be formally accepted for publication once it meets all outstanding technical requirements.

Kind regards,

Kai Wang

Academic Editor

PLOS ONE

Additional Editor Comments (optional):

Reviewers' comments:

Reviewer's Responses to Questions

**Comments to the Author**

1. If the authors have adequately addressed your comments raised in a previous round of review and you feel that this manuscript is now acceptable for publication, you may indicate that here to bypass the “Comments to the Author” section, enter your conflict of interest statement in the “Confidential to Editor” section, and submit your "Accept" recommendation.

Reviewer #1: All comments have been addressed

2. Is the manuscript technically sound, and do the data support the conclusions?

Reviewer #1: Yes

3. Has the statistical analysis been performed appropriately and rigorously? 

Reviewer #1: I Don't Know

4. Have the authors made all data underlying the findings in their manuscript fully available?

Reviewer #1: Yes

5. Is the manuscript presented in an intelligible fashion and written in standard English?

Reviewer #1: Yes

6. Review Comments to the Author

Reviewer #1: All comments raised during the review process have been thoroughly and appropriately addressed by the author, with necessary revisions and clarifications incorporated into the manuscript.

7. PLOS authors have the option to publish the peer review history of their article (what does this mean?). If published, this will include your full peer review and any attached files.

Reviewer #1: **Yes: **Fahrul Zaman Huyop

---

## [Editor Report · Acceptance letter]

PONE-D-24-51855R2

PLOS ONE

Dear Dr. Gómez Jaramillo,

I'm pleased to inform you that your manuscript has been deemed suitable for publication in PLOS ONE. Congratulations! Your manuscript is now being handed over to our production team.

Kind regards,

on behalf of

Dr. Kai Wang

Academic Editor

PLOS ONE